# Mechanistic molecular responses of the giant clam *Tridacna crocea* to *Vibrio coralliilyticus* challenge

**Duo Xu[1,2], Zehui Zhao[1,2], Zihua Zhou[1,2], Yue Lin[1,2], Xiangyu Zhang[1,2], Yang Zhang[1,3,4], Yuehuan Zhang[1,3,4], Jun Ii[1,3,4], Fan Mao[1,3,4], Shu Xiao[1,3,4], Haitao Ma[1,3,4], Xiang Zhiming** [1,3,4]*, **Ziniu Yu[1,3,4]***

**1** Key Laboratory of Tropical Marine Bio-resources and Ecology, Guangdong Provincial Key Laboratory of Applied Marine Biology, Chinese Academy of Science, South China Sea Institute of Oceanology, Guangzhou, China, **2** University of Chinese Academy of Sciences, Beijing, China, **3** Southern Marine Science and Engineering Guangdong Laboratory, Guangzhou, China, **4** Innovation Academy of South China Sea Ecology and Environmental Engineering (ISEE), Chinese Academy of Sciences, Guangzhou, China

* carlzyu@scsio.ac.cn (ZNY); zhimingxiang@scsio.ac.cn (ZMX)

**Data Availability Statement:** All sequencing reads were stored into the Short Read Archive (SRA) database and available with the accession PRJNA596951. Gene sequence is available from

## Abstract

*Vibrio coralliilyticus* is a pathogen of coral and mollusk, contributing to dramatic losses worldwide. In our study, we found that *V. coralliilyticus* challenge could directly affect adult *Tridacna crocea* survival; there were dead individuals appearing at 6 h post infection, and there were 45.56% and 56.78% mortality rates in challenged groups after 36 h of infection. The apoptosis rate of hemocytes was significantly increased by 1.8-fold at 6 h after *V. coralliilyticus* injection. To shed light on the mechanistic molecular responses of *T. crocea* to *V. coralliilyticus* infection, we used transcriptome sequencing analysis and other relevant techniques to analyze *T. crocea* hemocytes at 0 h, 6 h, 12 h and 24 h after *V. coralliilyticus* challenge. Our results revealed that the total numbers of unigenes and DEGs were 195651 and 3446, respectively. Additional details were found by KEGG pathway enrichment analysis, where DEGs were significantly enriched in immune-related signaling pathways, such as the TLR signaling pathway, and some were associated with signaling related to apoptosis. Quantitative validation results illustrated that with exposure to *V. coralliilyticus*, the expression of TLR pathway members, TLR, MyD88, IRAK4, TRAF6, and IκB-α, were significantly upregulated (by 22.9-, 9.6-, 4.0-, 3.6-, and 3.9-fold, respectively) at 6 h. The cytokine-related gene IL-17 exhibited an increase of 6.3-fold and 10.5-fold at 3 h and 6 h, respectively. The apoptosis-related gene IAP1 was dramatically increased by 2.99-fold at 6 h. These results indicate that adult *T. crocea* could initiate the TLR pathway to resist *V. coralliilyticus*, which promotes the release of inflammatory factors such as IL-17 and leads to the activation of a series of outcomes, such as apoptosis. The response mechanism is related to the *T. crocea* immunoreaction stimulated by *V. coralliilyticus*, providing a theoretical basis for understanding *T. crocea* immune response mechanisms.

the NCBI database (accession number MN829819).

**Funding:** Support was provided by the Strategic Priority Research Program of the Chinese Academy of Sciences, Grant No. XDA13020403/XDA13020202, the National Key R&D Program of China (2018YFC1406505), the National Natural Science Foundation of China (No. 31572661), Science and Technology Planning Project of Guangzhou, China (201707010177), Key Special Project for Introduced Talents Team of Southern Marine Science and Engineering Guangdong Laboratory (Guangzhou) (GML2019ZD0407), Institution of South China Sea Ecology and Environmental Engineering, Chinese Academy of Sciences (No. ISEE2018PY01) the National Science Foundation of China (31702340), Science and Technology Planning Project of Guangzhou, China (2017B030314052). The funders had no role in study design, data collection and analysis, decision to publish, or preparation of the manuscript.

**Competing interests:** The authors have declared that no competing interests exist.

## Introduction

The giant clam *Tridacna crocea*, inhabiting many Indo-Pacific coral reef communities, is remarkable for its fantastic mantle and unique ability to bore fully into coral rock [1]. Similar to other tridacnids, there is a symbiotic relationship between *T. crocea* and symbiotic dinoflagellates, which are commonly called zooxanthellae. Symbionts of clams are intercellular, and the zooxanthellae live within a branched, tubular structure permeating the mantle [2];, the symbionts produce nutrients via photosynthesis to aid in the host's autotrophy, while they gain essential nutrients for growth and metabolism from the host [3]. Because of the characteristics of giant clams, they are effective ecosystem engineers playing multiple roles in coral reefs. Nevertheless, because of environmental and anthropogenic disturbances, such as over-harvesting and habitat destruction, giant clam populations have been depleted, and their densities have been insufficient for self-replenishing and maintaining their populations, which might eventually lead to their extinction [4–6].

In recent years, the breeding of giant clams has attracted increasing attention worldwide. However, diseases caused by bacterial pathogens have been reported to result in losses in aquaculture populations of these commercially important shellfish [7]. Pathogenic vibrios can affect larval stages of cultured bivalves and are also involved in diseases of juveniles and adults [8]. *Vibrio coralliilyticus* was originally known as a temperature-dependent etiological agent involved in coral bleaching; it targets the mucus of the coral host through the use of dimethylsulfoniopropionate as a chemotaxic and chemokinetic agent, and its use of extracellular proteases results in coral tissue lysis and symbiont density decrease [9–11]. Lately, it was demonstrated to cause mortalities in larval oysters [12]. Since *V. coralliilyticus* is phylogenetically related to *Vibrio tubiashii*, many marine isolates of *V. coralliilyticus* from shellfish were misidentified as *V. tubiashii* [9, 13]. Therefore, previous reports of *V. tubiashii* in bivalve shellfish aquaculture on the west coast of North America were possibly caused by *V. coralliilyticus* [12, 14]. It has been reported that gross pathological changes to the velum and cilia occurred in diseased *Crassostrea gigas* larvae [15]. Moreover, *V. coralliilyticus* has also been associated with outbreaks of vibriosis in several other bivalve species, including Eastern oysters (*Crassostrea virginica*), the European flat oyster (*Ostrea edulis*), the great scallop (*Pecten maximus*), the Atlantic bay scallop (*Argopecten irradians*) and New Zealand green-lipped mussels (*Perna canaliculus*) [16, 17]. Consequently, *V. coralliilyticus* is a significant pathogen for bivalves, contributing to dramatic losses in mollusks worldwide.

Bivalves mostly rely on an innate immune system composed of cellular and humoral components [18]. Hemocytes are thought to be responsible for the main activities by which bivalves respond to infectious agents, and those are phagocytosis, encapsulation and nacrezation [19]. The first action of the bivalve's immune system when challenged by microbial pathogens is the recognition of these foreign organisms. This is achieved by expressing various pattern recognition receptors (PRRs), which sense diverse pathogen-associated molecular patterns (PAMPs). Recognition of PAMPs by PRRs activates intracellular signaling pathways that culminate in the induction of inflammatory cytokines, chemokines, interferons (IFNs) and the upregulation of costimulatory molecules [20]. Toll-like receptors (TLRs) are particularly essential members of PRRs, and they are responsible for the recognition of PAMPs and the activation of downstream signaling adaptors [21]. Among these adaptors, myeloid differentiation factor 88 (MyD88), which mediates the activation of the TLRs, except for TLR3, was originally identified as a curial and conserved signaling proteins [22, 23]. MyD88 interacts with IL-1 receptor-associated kinase (IRAK), and IRAK associates with TNF receptor-associated factor 6 (TRAF6), which subsequently activates the NF-κB pathway [24]. In addition, the TLR pathway not only activates inflammation and phagocytosis but also regulates the induction of apoptosis

[25]. Apoptosis is important in the immune system and plays significant roles in the control of the immune response, the removal of immune cells recognizing self-antigens, and cytotoxic killing [26]. Pathogens can be eliminated by the ability of hemocytes to recognize foreign targets and induce apoptosis [27]. Similarly, *V. coralliilyticus* exposure can induce significant changes in the host TLR pathway and apoptotic systems [28, 29]. Since the first TLR was identified in *Drosophila melanogaster*, large members of TLR family have been recently investigated in marine bivalves, such as *Chlamys farreri*, *Chlamys nobilis*, *Mizuhopecten yessoensis*, *C. gigas*, *C. virginica*, *Mytilus edulis* and *Hyriopsis cumingii* [30–36]. Among them, the innate immune regulation of TLR genes in bivalves has been reported in *C. farrer*i (*Cf*Toll-1) and *C. nobilis* (*Cn*TLR-1), respectively, both of which might be involved in the immune response against pathogen invasion [31, 37]. In addition, 23 TLRs were identified and arranged in 4 clusters according to extra-cellular LRR domain content in *Mytilus galloprovincialis* [38]. Furthermore, there were 83 TLR genes in the genome of *C. gigas*, 19 of which had different responses to *Vibrio* infection [39]. Despite the fact that *T. crocea* has high ecological value and is under the stress of populations, there is limited information available about its immune molecule response mechanisms.

In the present study, we injected adult *T. crocea* with *V. coralliilyticus* to investigate the individual and cellular responses of giant clams. Furthermore, high-throughput sequencing was applied to analyze the differentially expressed genes in *T. crocea* hemocytes at 0 h, 6 h, 12 h, and 24 h after *V. coralliilyticus* challenge. Afterwards, using KEGG pathway enrichment analysis, some molecular mechanisms of response and candidate genes involved in *V. coralliilyticus* infection were identified. Therefore, we explored the possible sensing patterns for *V. coralliilyticus* in the innate immune system of *T. crocea*. The results provide insights for a better understanding of *V. coralliilyticus* pathogenicity and the future development of disease prevention strategies.

## Materials and methods

### Ethics statement

All *T. crocea* used in the present study were bred in our laboratory (Hainan Tropical Marine Biology Research Station), Chinese Academy of Sciences, Sanya, China. No specific permits were required for clams sample collection or described sampling. The location was not privately-owned or protected, and the field studies did not involve any endangered or protected species.

### Animals, tissue, embryonic development collection and challenge experiment

Adult *T. crocea* (average 6.7–10.8 cm shell length) were obtained from Sanya, Hainan Province, China and maintained in tanks filled with natural seawater (temperature: 28±1˚C; salinity: 33). Cultures were illuminated with metal halide bulbs from 6:00 am to 7:00 pm every day for two weeks prior to use.

To analyze the tissue distribution of *Tc*MyD88, the following tissues were collected from six healthy adult *T. crocea*: siphonal mantle, pedal mantle, inner mantle, byssus gland, pedis, heart, gill and hemocytes. For analysis of the developmental expression patterns of *Tc*MyD88, samples of 0, 0.5, 1, 4, 12, 16, 24, 48, 72, 96, and 120 h-developing embryos were also obtained.

For the bacterial challenge, an in vivo infection experiment was performed. Healthy giant clams were injected into the adductor muscle either 100 μL of 2×PBS or 100 μL of live *V. coralliilyticus* ($1.0 \times 10^8$ CFU/mL) suspended in 2×PBS. Specifically, purified bacterial inoculum

were centrifuged for 10 min to collect bacterial pullets and were washed 3 times with PBS and re-suspended in 2×PBS to a concentration of OD600nm = 1.0. *V. coralliilyticus* (CAIM616) was purchased from the Marine Culture Collection of China and was cultured in Zobell marine broth 2216 (Difco Laboratories) at 28 ˚C overnight. After injection, the giant clams were returned to tanks full of seawater at 28 ˚C for subsequent treatment or sampling. If the valves of a giant clam were not closed and the mantle did not react after stimulation, it was considered dead. Mortality in each tank was assessed visually and counted every 6 h after injection. Hemocytes were collected at scheduled intervals (0, 3, 6, 12, 24, and 36 h post injection) from both the challenged and control groups. Among them, hemocytes taken at 0, 6, 12, and 24 h post challenge were stored in liquid nitrogen for transcriptome analysis. Five individuals were randomly sampled from each group at every time point after injection to obtain biological replicates.

## Flow cytometric analysis of apoptosis

Hemocytes were harvested 6 h after *V. coralliilyticus* injection and were then resuspended in 100 μL of binding buffer containing 5 μL of Annexin V-FITC and 5 μL of propidium iodide, provided in an apoptosis detection kit (Vazyme, A211); the mix was incubated at room temperature for 10 min in the dark. Finally, another 400 μL of binding buffer was added to the solution and applied to a Guava® easyCyte™ (Millipore). At least 10,000 cells were obtained to analyze the population by Flow Jo v10.0 software.

## De novo assembly and gene annotation

Based on the sequencing by synthesis (SBS) technique, cDNA libraries were sequenced using an Illumina Hiseq high-throughput metering platform to obtain raw data. The raw data were processed to discard the dirty reads and low-quality sequences. After filtering, the remaining reads were called "Clean Reads". The unigenes were de novo assembled using Trinity [40]. For gene annotation analysis, the assembled transcripts were scanned against NR (NCBI nonredundant protein sequences), Swiss-Prot databases and KOG (euKaryotic Ortholog Groups) using diamond with E-values at $1.0 \times 10^{-5}$ (E-values of less than $1.0 \times 10^{-5}$ were considered significant). NT (NCBI nucleotide sequences) and PFAM (protein family) analyses were performed using the NCBI BLAST + v2.2.28 and HMMER 3.0 programs, respectively. In addition, the unigenes were also classified according to the GO (Gene Ontology) and KEGG (Kyoto Encyclopedia of Genes and Genomes) databases.

## Differential expression analysis

Read counts were estimated by mapping clean reads to unigenes using Bowtie2, and they were calculated according to the comparison result with RSEM [41]. The expression abundance of the corresponding Unigene was expressed by the FPKM (expected number of fragments per kilobase of transcript sequence per millions of base pairs sequenced) value [42]. We used the adjusted p-value to detect differentially expressed genes (DEGs). When the adjusted p-value was less than 0.05 and had a greater than two-fold change (absolute value of log2 ratio≥1), the gene was considered differentially expressed in a given library. Significantly enriched terms were obtained by mapping DEGs to the corresponding KEGG pathways.

## RNA extraction and cDNA synthesis

Total RNA (50mg) was isolated with TRIzol Reagent (Invitrogen, 15596–026) according to the manufacturer's protocol. RNA was dissolved in DEPC-treated water, and the integrity of the

RNA was assessed by electrophoresis with a 1.0% agarose gel, and then the concentration and purity were examined at 260/230 and 260/280 absorbance ratios. Purified RNA was diluted to 1 mg/ml to synthesize first-strand cDNA using a Primer Script™ First Strand cDNA Synthesis kit (TAKARA Bio Inc. Japan). The cDNA was used as the template for amplifying gene sequences and analyzing their expression. All primers used in this study were designed with Primer Premier 5.0 and are shown in S1 Table.

## Cloning the full-length cDNA of *Tc*MyD88

A search of the transcriptome data of *T. crocea* revealed a TIR contig homologous to the MyD88 gene of *T. crocea*. The intermediate fragment sequences of *Tc*MyD88 were obtained by polymerase chain reaction (PCR). Then, gene-specific primers were designed to amplify the unknown 5' and 3' ends of *Tc*MyD88 cDNA using rapid amplification of cDNA ends (RACE). For *Tc*MyD88 3' sequencing, the primer pairs Takara3P/*Tc*MyD88-F1 and Takara3NP/*Tc*MyD88-F2 were employed for primary PCR and nested PCR, respectively. Similarly, the 5' end of the *Tc*MyD88 gene was obtained by nested PCR using Takara5P/*Tc*MyD88-R1 and Takara5NP/*Tc*MyD88-R2. Full-length cDNA sequences were obtained by combining intermediate fragment sequences and 3' and 5' end sequences.

## Sequence and phylogenetic analyses

The cDNA sequences and deduced amino acid sequences of *Tc*MyD88 were analyzed using the BLAST program (http://blast.ncbi.nlm.nih.gov/Blast.cgi) and the Expert Protein Analysis System (http://expasy.org/). The nucleotide and protein sequences were analyzed using BLASTN and BLASTX, respectively. The molecular weights and the theoretical isoelectric points were calculated using the Compute pI/Mw tool (http://web.expasy.org/compute_pi/). A structural analysis of proteins was performed using the Simple Modular Architecture Research Tool (SMART) program (http://smart.embl-heidelberg.de/). Multiple protein sequence alignments were performed using Bio-edit software via the Clustal W method. A phylogenetic tree was constructed using the MEGA5.0 software based on the alignment of the complete amino acid sequences with the neighbor-joining method and 1000 bootstrap replicates.

## Plasmid construction, cell culture and transfection

The *Tc*MyD88 ORF was amplified from *T. crocea* cDNA by PCR using specific primers designed by CE Design V1.03. The target PCR products were subcloned into eukaryotic expression vectors, pcDNA3.1-HIS (Promega, USA) and pEGFP-N1 (Promega, USA), by homologous recombination using a Vazyme™ one step cloning kit. The constructed recombinant plasmids were digested with enzymes, and the inserted fragments of each clone were sequenced. All of the plasmids used for transfection were extracted from overnight bacterial cultures using a HiPure Plasmid EF Micro kit (Magen) according to the manufacturer's protocol.

HEK293T cells were cultured in Dulbecco's modified Eagle's medium (DMEM, Gibco) supplemented with 10% heat-inactivated fetal bovine serum (FBS) and antibiotics (100 mg/L streptomycin and 105 U/L penicillin, Gibco) at 37 ˚C with 5% $CO_2$, and they were subcultured every three days. Transfections were performed using ViaFect reagent (Promega, USA) according to the manufacturer's instructions.

## Subcellular localization and dual-luciferase reporter assays

For subcellular localization analysis, HEK293T cells were seeded into a 6-well cell plate, cultured for 24 h and then transfected with a pEGFP-N1-*Tc*MyD88 mix. Forty-eight hours after

transfection, cells were washed once with 1×PBS (pH 7.4) and then were fixed with 4% para-formaldehyde for 10 min, which was followed by staining with 6-diamidino-2-phenyl-indole (DAPI) (1 mg/ml) for 5 min. Finally, the cells transfected with fluorescent vectors were directly observed by fluorescence microscopy.

For dual-luciferase reporter assays, cells were plated in a 48-well cell plate, cultured until they reached 40–50% confluency and then were transfected with pRL-TK (20 ng/well), NF-κB reporter (200 ng/well) and a target plasmid (0, 100, 200 or 400 ng/well). The pRL-TK vector and NF-κB vector (Promega, USA) were used as internal controls. After 48 h of transfection, the luciferase activity of total cell lysates was measured using a luciferase reporter assay system (Promega, USA). Renilla luciferase activity was expressed as the fold stimulation relative to that of the empty vector transfected cells. The values are expressed as the mean relative stimu-lation for a representative experiment from four separate experiments, and each experiment was performed in duplicate.

### Quantitative real-time PCR analysis

qRT-PCR reactions were conducted using 2 × Real Star Green Power Mixture (GenStar, A311) and a LightCycler® 480 II (Roche, Switzerland) according to the manufacturer's protocol. All the template cDNA were diluted to 200ng/μl. PCR conditions were as follows: 95 ˚C for 10 min, followed by 40 cycles of amplification at 95 ˚C for 15 s, 55 ˚C for 30 s and 72 ˚C for 30 s. At the end of each qPCR, a melting curve analysis was performed to confirm the specificity of the PCR products. The data from each experiment are expressed relative to the expression levels of the β-actin gene to normalize expression levels between the samples. All of the samples were ana-lyzed in triplicate, and the expression values were calculated with the $2^{-\Delta\Delta CT}$ method [43].

### Statistical analysis

Data processing and statistical analyses were performed using GraphPad Prism v5.0.1. All data are presented as the means ± S.D. Comparisons between two groups of samples were per-formed with Student's t-tests, and comparisons among more than two groups were performed with one-way ANOVA followed by Turkey's or Dunn's post hoc test. Values *p < 0.05, **p < 0.01 and ***p < 0.001 were considered significant.

## Results

### The mortality and apoptosis after vibrio challenge

To investigate the possible effects of a *V. coralliilyticus* challenge on *T. crocea*, challenge experi-ment was performed. After injection with *V. coralliilyticus*, the siphonal mantle of *T. crocea* contracted inward and collapsed (Fig 1A). Bacterial challenge directly affected adult survival; dead individuals appeared at 6 h post infection, and 45.56% and 56.78% mortality rates were observed in the two challenged groups, while no death was observed in the control group after 36 h (Fig 1B). Flow cytometry was performed to detect the apoptosis of hemocytes. The results showed that the apoptosis rate of the hemocytes dramatically increased 1.8-fold (including early and late apoptotic cells) in the *V. coralliilyticus*-injected *T. crocea* compared with that of the control group, which clearly implied that *V. coralliilyticus* could cause apoptosis in *T. cro-cea* hemocytes (Fig 1C and 1D).

### Transcriptome sequencing and assembly of transcripts

The transcripts of *T. crocea* injected with *V. coralliilyticus* were assembled to analyze the molecular response mechanism of *T. crocea* during *V. coralliilyticus* infection. After filtering

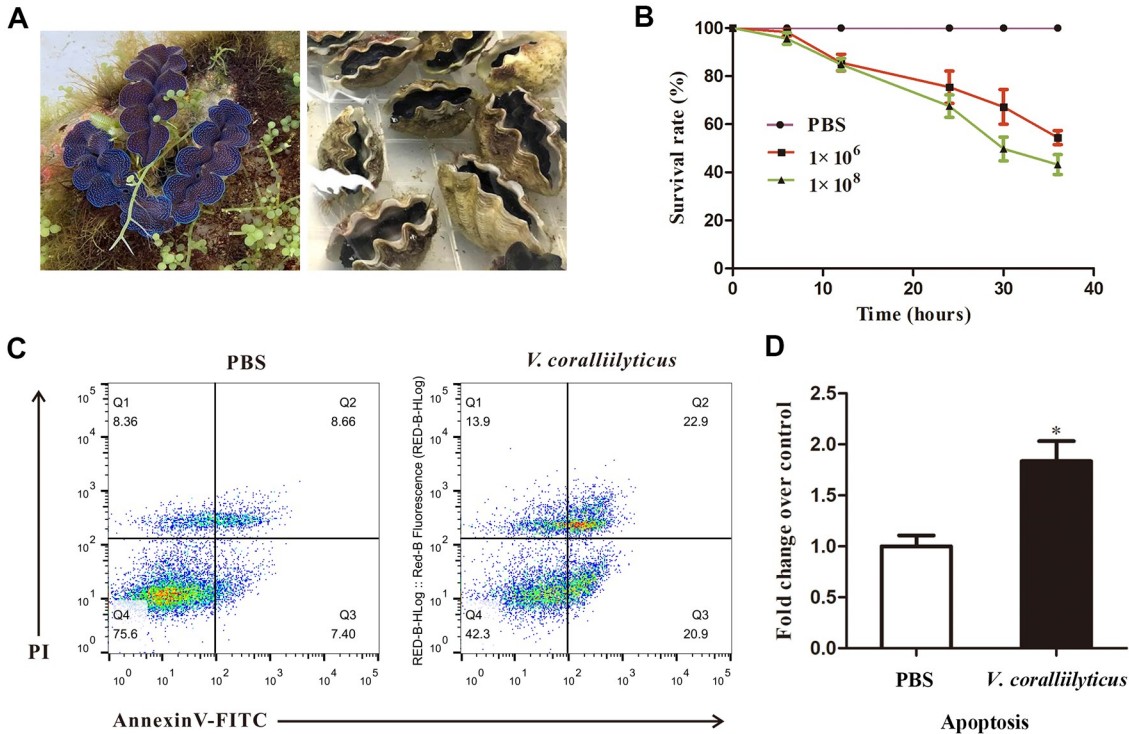

**Fig 1. *V. coralliilyticus* directly affected adult *T. crocea* survival and significantly promoted apoptosis in *T. crocea* hemocytes.**
(A) Healthy and infected adult *T. crocea*. (B) Survival rate of adult *T. crocea* as a function of the duration of the experiment and the varying concentrations of *V. coralliilyticus*. (C) To detect apoptosis of hemocytes after *T. crocea* was stimulated by *V. coralliilyticus*, flow cytometry was used with Annexin V-FITC and PI; $n = 3$. On the scatter plot of the bivariate flow cytometry, the left lower region represents the living cells, the lower right region represents the early apoptotic cells and the top right region represents the late apoptotic cells. (D) The apoptosis fold change following *V. coralliilyticus* challenge was compared to the control. Significant differences are indicated by an asterisk.

dirty reads from the raw reads, 46525916, 44448554, 49647448 and 53853546 clean reads were obtained from hemocytes in the *V. coralliilyticus* challenged group at 0 h, 6 h, 12 h, and 24 h, respectively. Sequencing reads were found to be of high quality, with Q30% values of 92.97%, 92.33%, 91.89% and 93.03%, respectively. The GC content of nucleotides was approximately 38%. For all four sequencing libraries, the percentages of reads that could be matched with assembled reference sequences were higher than 64.80% (Table 1). All sequencing reads were stored into the Short Read Archive (SRA) database and available with the accession PRJNA596951.

## Annotation and function analysis of unigenes

To achieve protein identification and gene annotation, transcripts were compared with data from the NR, NT, KO, Swiss-Prot, PFAM, GO and KOG databases. A total of 195651 unigenes

**Table 1. Quality of sequencing data.**

| Sample | Clean Reads | Clean Bases | Q30 (%) | GC (%) | Mapped Reads | Mapped Ratio (%) |
|---|---|---|---|---|---|---|
| V0 | 46525916 | 6.98G | 92.97 | 38.35 | 30622674 | 65.82% |
| V6 | 44448554 | 6.67G | 92.33 | 38.26 | 29358866 | 66.05% |
| V12 | 49647448 | 7.45G | 91.89 | 38.28 | 32169732 | 64.80% |
| V24 | 53853546 | 8.08G | 93.03 | 38.28 | 34899862 | 64.81% |

(ranging from 301 to 25533 bp) were generated with an N50 size of 810 bp (Table 2). All sequences from 75468 unigenes were annotated in at least one database, with 28385 annotated unigenes (14.50%) having significant matches with sequences in the NR. We also assembled the species distribution of the unigenes by aligning sequences against the NR database to learn the sequence similarity of *T. crocea* with other species. A total of 62.5% of the unigenes matched with sequences from five top-hit species, *M. yessoensis* (25.4%), *C. gigas* (14.3%), *C. virginica* (14.0%), *Lottia gigantea* (5.2%) and *Lingula anatina* (3.6%), all of which are mollusks (S1 Fig).

GO classification is a unified gene functional classification system. The analysis of annotated transcripts demonstrated that 36361 matched unigenes (18.58%) were divided into 26, 20, 10 entries for three categories, biological processes (BP), cellular component (CC) and molecular function (MF), respectively. The unigenes in "binding," "catalytic activity," "signaling," "response to stimulus," "membrane," which were potentially related to immune existed in higher percentages compared to counterparts at sub-categories at the whole transcriptome reference. They played important roles during the defense process after challenge. The results provided a comprehensive view for screening candidate genes related to immune and defense mechanism (S2 Fig).

In addition, we also used KEGG to identify the probable functional status of assembled transcripts. A total of 12912 unigenes (6.59%) were assigned to five main categories containing 232 KEGG pathways. Remarkably, immune system (558) was the well-represented term in the KEGG organismal systems category. Signal transduction (1455) was the most enriched terms in both environmental information processing and the whole transcriptome reference, suggesting the dominant position of various signal transduction in processing the stimulation of environment. In the metabolism category, abundant unigenes were found in the amino acid metabolism (510) sub-category. These results indicated that these unigenes may play a role in mediating *V. coralliilyticus* exposure and associated impacts (S3 Fig).

## Identification of differentially expressed genes

To identify the DEGs involved in *V. coralliilyticus* infection, a comparison of the relative transcript abundance for each unigene was performed. We compared the expression levels of each unigene for 6 h, 12 h, and 24 h with 0 h, obtaining 1941, 2172 and 1116 DEGs, respectively (Fig 2A). By taking the union of three sets of DEGs, we ended up with 3446 DEGs (Fig 2B and S2 Table). Subsequently, KEGG pathway enrichment analysis was used to classify the DEGs and

**Table 2. Summary of annotations of all assembled unigenes.**

|  | Number of unigenes | Percentage (%) |
| --- | --- | --- |
| Total number of unigenes | 195651 | |
| Mean length of unigenes (bp) | 739 | |
| N50 length of unigenes (bp) | 810 | |
| Max length of unigenes (bp) | 25533 | |
| Min length of unigenes (bp) | 301 | |
| Annotated in NR | 28385 | 14.50 |
| Annotated in NT | 40147 | 20.51 |
| Annotated in KO | 12912 | 6.59 |
| Annotated in SwissProt | 20653 | 10.55 |
| Annotated in PFAM | 36361 | 18.58 |
| Annotated in GO | 36361 | 18.58 |
| Annotated in KOG | 11768 | 6.01 |
| Annotated in at least one Database | 75468 | 38.57 |

highlight biological associations. As expected, the results emphasized the immune system, where the Toll-like receptor signaling pathway and the NF-κB signaling pathway were enriched in DEGs. Furthermore, some DEGs were associated with apoptotic signaling, such as the TNF signaling pathway (Fig 3A). Meanwhile, we selected some of the DEGs related to innate immunity and constructed a heat map in which 39 genes were upregulated and 16 genes were downregulated (S3 Table). The upregulated genes were mainly TLR pathway-related genes, immune effector molecules (which included inflammatory cytokines interleukin-1 (IL-1), IL-17 and tumor necrosis factor (TNF)), antimicrobial peptides (β defensin and phage lysozyme 2), and apoptosis-related genes (such as inhibitor of apoptosis proteins 1 (IAP1) and the anti-apoptotic genes of the Bcl-2 family). For the TLR pathway, one TLR homolog, the adaptor molecule MyD88, TRAF6, kinase IRAK4 and two transcription factors IκB-α and NF-κB were all significantly upregulated after 6 h and 12 h of infection. The downregulated genes included members of the lectin family and the complement pathway, including perlucin, galectin, and C-type lectin (Fig 3B). The results revealed that the MyD88-dependent pathway was activated after *V. coralliilyticus* challenge, which promoted the release of immune effector molecules such as IL-1 and TNF, and indirectly affected the expression of some apoptosis-related genes.

## Cloning and sequence analyses of *Tc*MyD88

*Tc*MyD88 was dramatically enriched in multiple KEGG pathways and occupied an essential position in the TLR pathway. The full-length sequence of *Tc*MyD88 (MN829819) is 1853 bp, containing a 105 bp 5'-untranslated region (UTR), a 140 bp 3'-UTR and an open reading frame (ORF) of 1608 bp. The ORF encodes a putative protein of 535 amino acids, with an estimated molecular mass of 60.430 kDa and a theoretical isoelectric point (PI) of 5.04 (S4 Fig).

The protein structure of *Tc*MyD88 was predicted and analyzed using the SMART program. Typical Myd88 domains comprising a DEATH domain (residues 24–123) and a TIR domain (residues 179–316) were identified in the *Tc*MyD88 protein (Fig 4A).

To clarify the evolutionary relationship among variable MyD88 proteins, multiple sequence alignments were performed with the deduced amino acid sequences of MyD88. Compared with other species, *Tc*MyD88 displayed a high degree of homology and conservation, especially in the TIR domain (Fig 4B). Moreover, a phylogenetic tree was also constructed using MEGA5.0 software using the neighbor-joining method. The deduced phylogeny of MyD88 revealed two

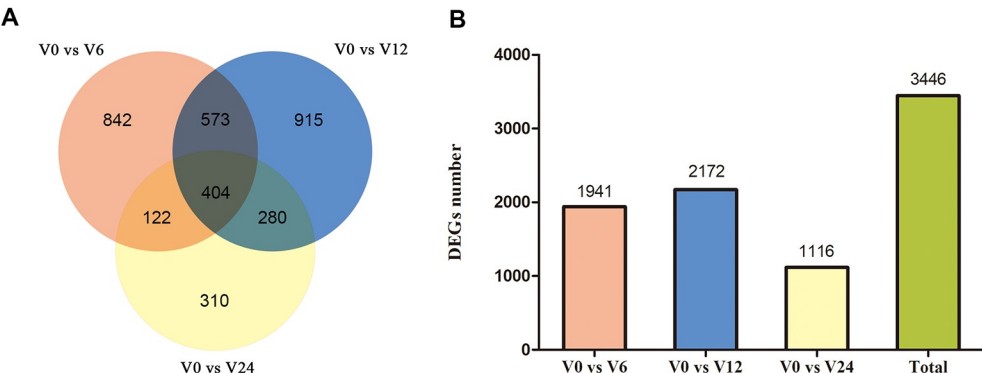

**Fig 2. Differentially expressed genes in the *V. coralliilyticus*-infected samples compared to sample V0.** (A) Venn diagram visualizes overlapping transcripts between V0 vs. V6, V0 vs. V12 and V0 vs. V24 transcriptomes. The different circles represent comparisons of different samples. The numbers in the fields describe the number of transcripts that the three analyzed groups of DEGs have in common at a given intersection. (B) Numbers of DEGs in V0 vs. V6, V0 vs. V12, V0 vs. V24 and total.

major clusters, one for mollusks and the other for fish and mammals. *Tc*MyD88 initially clustered with bivalves, such as *Ruditapes philippinarum*, *H. cumingii*, *C. virginica*, and *C. giga*, to form a single branch. In conclusion, the evolution of *Tc*MyD88 is relatively well conserved in bivalves, and it is distantly related to that of vertebrates and other invertebrates (Fig 4C).

## Expression pattern variation of *Tc*MyD88 in tridacna

The tissue distributions of *Tc*MyD88 mRNA were detected by qRT-PCR. According to the results, *Tc*MyD88 was widely expressed in all examined tissues, including the siphonal mantle, pedal mantle, inner mantle, byssus gland, pedis, heart, gill and hemocytes. With the change in the position of mantles, the expression level of *Tc*MyD88 varied, while it was relatively highly expressed in the siphonal mantle. Furthermore, it was expressed predominantly in the gills, followed by the siphonal mantle and hemocytes, whereas the pedis, heart and byssus gland contained low levels. The expression level of *Tc*MyD88 was approximately 33.7-fold higher in gills than it was in hearts (Fig 5A).

The mRNA expression pattern of *Tc*MyD88 was also observed throughout developmental stages. As shown in the figure, the expression of *Tc*MyD88 maintained a stable, low level from 0 h to 72 h. At 72 h, there was a sudden increase, and the level continuously increased after 72 h (Fig 5B).

## Subcellular localization of *Tc*MyD88

The subcellular localization of *Tc*MyD88 was examined by transient transfection of the *Tc*MyD88-GFP plasmid into HEK293T cells. Imaging of the GFP-tagged *Tc*MyD88 revealed that *Tc*MyD88 was distributed mainly in the cytoplasm, whereas the GFP protein was found in both the cytoplasm and the nucleus (Fig 6).

## Dual-luciferase reporter assays

To determine whether *Tc*MyD88 could modulate NF-κB transcriptional activity, dual-luciferase reporter assays were performed in HEK293T cells. As shown in Fig 7, *Tc*MyD88 can activate the NF-κB responsive reporter, and the overexpression significantly increased the activation in a dose-dependent manner from 100 ng to 400 ng. The most marked increase was approximately 10.9-fold (P < 0.01) over what was observed in cells transfected with pCDNA3.1 alone. These results implied that *Tc*MyD88 could potentially be involved in the NF-κB signaling pathway.

## The responses of TLR pathway-related genes in *T. crocea* to *V. coralliilyticus* challenge

On account of the DEG analysis showing that the TLR pathway was activated, we proceeded to analyze pathway-related genes by qRT-PCR. For the TLR pathway genes, significant differences between control and infected *T. crocea* individuals were observed for TLR, MyD88, IRAK4, TRAF6 and IκB-α. Upon *V. coralliilyticus* challenge, the level of TLR transcripts was significantly upregulated at 6 h (22.9-fold; p < 0.01) and then returned to the basal level at 36 h (Fig 8A). With *V. coralliilyticus* stimulation, the expression of *Tc*MyD88 initially significantly increased at 3 h (4.8-fold; p < 0.01) and reached the highest expression levels at 6 h (9.6-fold; p < 0.01), and then it declined at 12 h (Fig 8B). During the *V. coralliilyticus* challenge, the expression level of IRAK4 was upregulated at 3 h and maintained a high level throughout the whole stage, with the highest value at 6 h (4.0-fold; p < 0.01); then it returned to the basal level at 36 h (Fig 8C). With exposure to *V. coralliilyticus*, TRAF6 expression was significantly upregulated by 3.6-fold (p < 0.01) at 6 h, and then it peaked with a 2.8-fold increase at 12 h

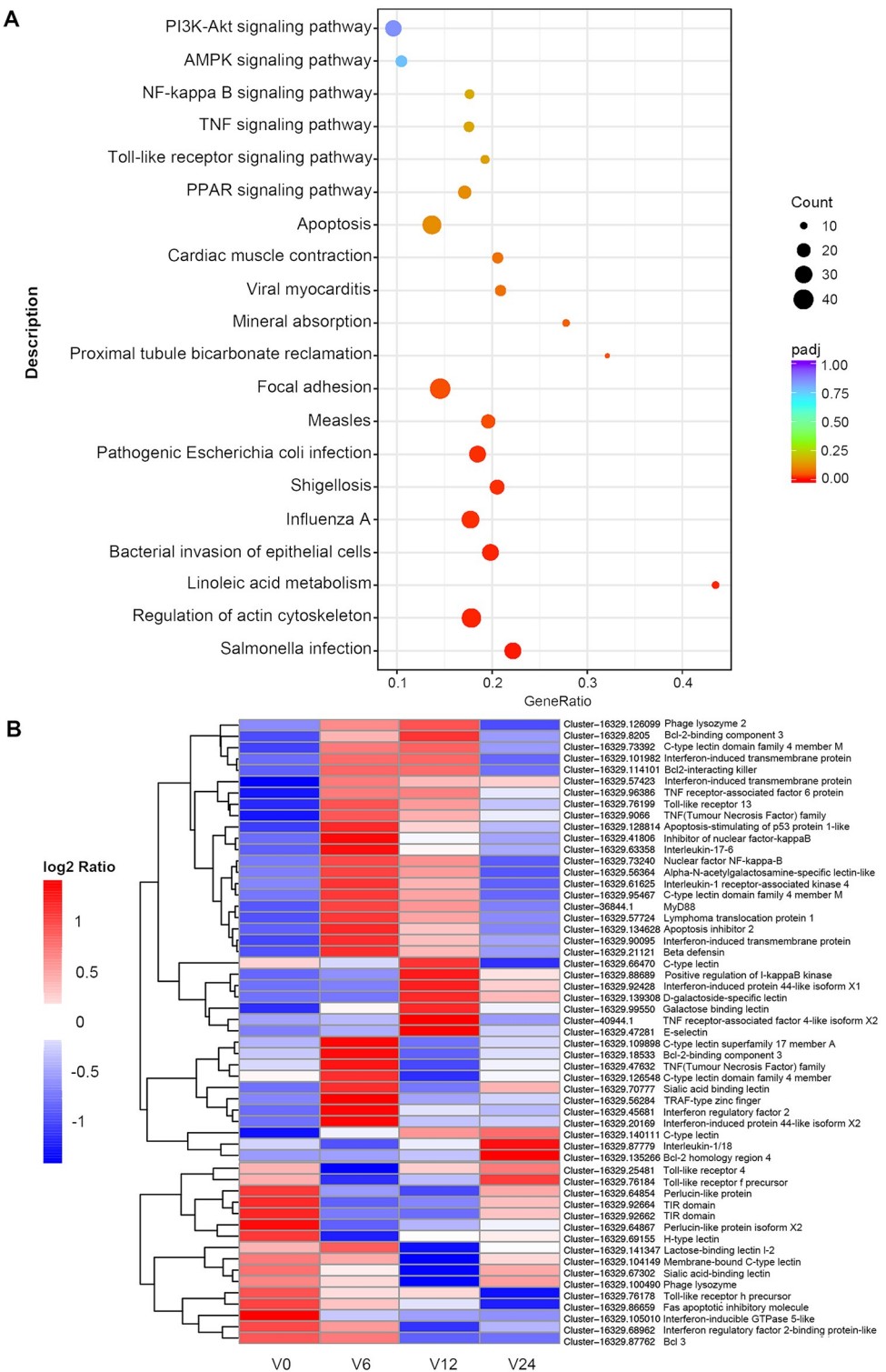

**Fig 3. Differentially expressed genes analysis.** (A) KEGG pathway enrichment analysis of DEGs in *T. crocea* hemocytes exposed to *V. coralliilyticus*. The enrichment factor is the ratio of the DEG number and the number of all genes in a certain enrichment pathway. The dot size denotes the number of DEGs, while colors correspond to the adjusted p-value range. (B) A heat map shows immune-related differentially expressed *T. crocea* transcripts. The heat map shows expression profiles of healthy *T. crocea* at 0 h, 6 h, 12 h and 24 h after *V. coralliilyticus* challenge. Color intensity is proportional to the magnitude of changes. Relative expression levels are shown in red (upregulation) and blue (downregulation).

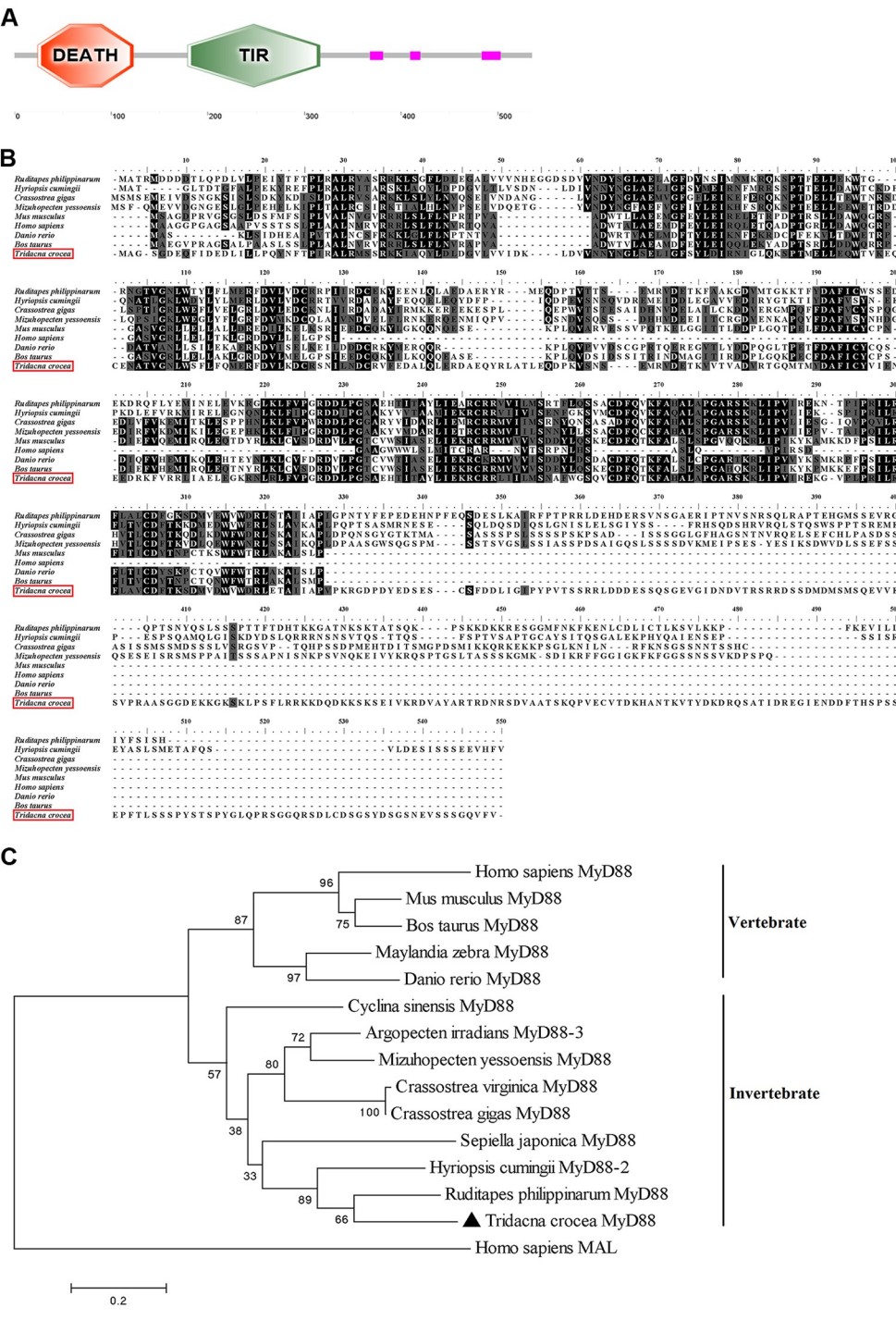

**Fig 4. Sequence analysis of *Tc*MyD88.** (A) Functional domain organization of *Tc*MyD88. (B) Multiple sequence alignment analysis of *Tc*MyD88 and other known members of the MyD88 family. The black shaded sequences indicate residues that exactly match the consensus. The gray shaded sequences showed a similar relationship. The GenBank accession numbers corresponding to the MyD88 sequences examined are listed: *Ruditapes philippinarum* (AEF32114.1), *Hyriopsis cumingii* (AHB62785.1), *Crassostrea gigas* (NP_001292286.1), *Mizuhopecten yessoensis* (XP_021355224.1), *Mus musculus* (NP_034981.1), *Homo sapiens* (NP_001166037.2), *Danio rerio* (NP_997979.2), and *Bos taurus* (NP_001014404.1). (C) A phylogenetic tree showing the relationships among *Tc*MyD88 and MyD88 from different species was constructed by the neighbor-joining method using MEGA5.0 software. The node values represent the percent bootstrap confidence derived from 1000 replicates. The GenBank accession numbers corresponding to the MyD88 sequences and MAL sequence examined: *Argopecten irradians* (AVP74319.1), *Crassostrea virginica*

(XP_022332088.1), *Cyclina sinensis* (AIZ97751.1), *Sepiella japonica* (AQY56781.1), *Maylandia zebra* (XP_004546813.1), *Homo sapiens* MAL (NP_002362.1), and other sequences referenced in Fig 4B.

(p < 0.001) (Fig 8D). After *V. coralliilyticus* infection, the expression of IκB-α rapidly increased at 6 h (3.9-fold; p < 0.001) and then dropped to 1.5-fold at 12 h (p < 0.05) before nearing the control level at 36 h (Fig 8E). The expression of cytokine-related genes was also affected by infection. In response to *V. coralliilyticus* challenge, IL-17 exhibited an increase at 3 h and 6 h by 6.3-fold (p < 0.001) and 10.5-fold (p < 0.001), respectively (Fig 8F). There was also a significant change in apoptosis-related gene IAP1. During *V. coralliilyticus* infection, IAP1 mRNA was increased at 3 h (2.0-fold; p < 0.01), and it dramatically increased at 6 h with a 3.0-fold change (p < 0.001) (Fig 8G). In summary, the qRT-PCR analyses greatly confirmed the direction of changes determined by the transcriptome analysis. Preliminary indications are that the expression of TLR pathway-related genes changed after *V. coralliilyticus* challenge, and the downstream genes were activated, which promoted the release of inflammatory and led to the activation of a series of signals.

## Discussion

In a global warming scenario, an increase in the seawater temperature could promote the potential disease outbreaks associated with *V. coralliilyticus* in mollusks [44]. In the present study, we demonstrated that *V. coralliilyticus* could cause acute mortalities of adult *T. crocea* when previous work was mainly focused on mortalities of shellfish larvae caused by *V. coralliilyticus* [12, 45]. After injection with *V. coralliilyticus*, the siphonal mantle of *T. crocea* contracted inward and collapsed, while the siphonal mantle of healthy *T. crocea* extended outward. In previous studies, *V. coralliilyticus* infection resulted in the loss of *C. gigas* larval motility due to the gross pathological changes to the velum and cilia [15]. Histopathology indicated that the route of infection by *V. coralliilyticus* was the digestive system in Greenshell™ mussel larvae. Furthermore, vacuolation of the tissues of the digestive tract similar to oyster *C. virginica* larvae and necrotic tissue were observed [45, 46]. Moreover, the results of this study suggest that *V. coralliilyticus* could trigger apoptosis in *T. crocea* hemocytes. Diverse metalloprotease and effector genes like the pore forming toxin hlyA were identified in the genome of

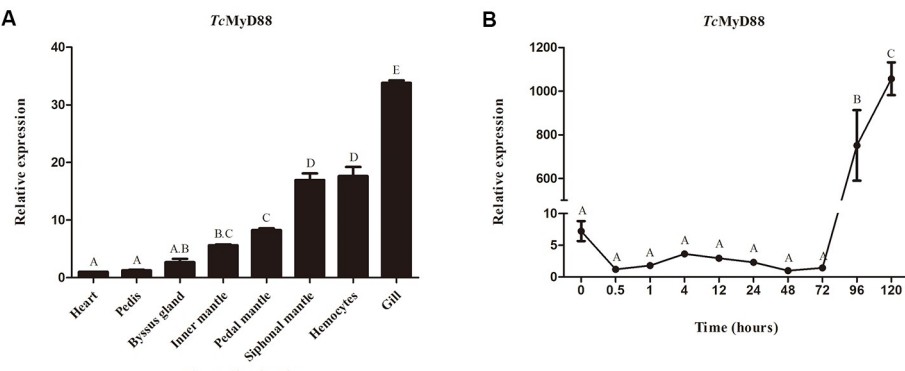

**Fig 5. Expression pattern of *Tc*MyD88.** (A) The expression patterns of *Tc*MyD88 in different tissues were examined by qRT-PCR. The relative expression levels were normalized to β-actin. Significant differences are indicated by different letters. (B) Relative expression levels of *Tc*MyD88 in different embryonic stages. β-actin was employed as an internal control. Error bars indicate standard error.

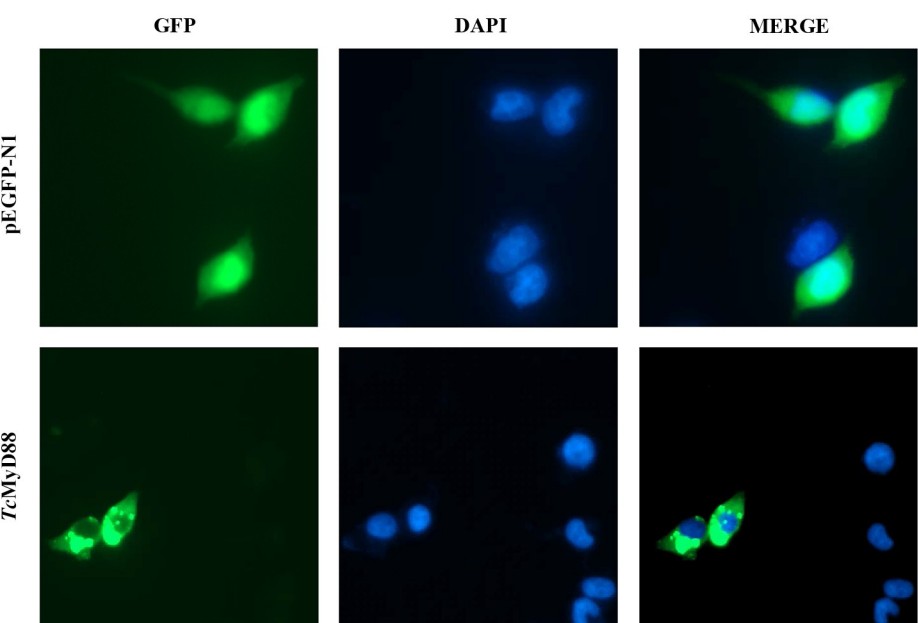

**Fig 6. Subcellular localization of *Tc*MyD88.** The left part of the panel shows the pEGFP-N1 and *Tc*MyD88 fluorescence fusion proteins, the middle of the panel shows the cell nuclei indicated by blue DAPI staining, and the right part of the panel shows the combined images.

*V. coralliilyticus* and expressed proteases were also detected in the secretome, which caused mortality in *Drosophila* and *Artemia* and may be involved in the infection of *T. crocea* [47]. It has been reported that hemolysins play a role in inducing apoptosis [48]. In addition, *C. gigas* larvae infected by *V. coralliilyticus* showed higher activities of catalase and superoxide dismutase, two key enzymes implicated in antioxidant defense, indicating their activation after pathogen stimulation; further, PO activity was significantly increased in challenged mussel larvae

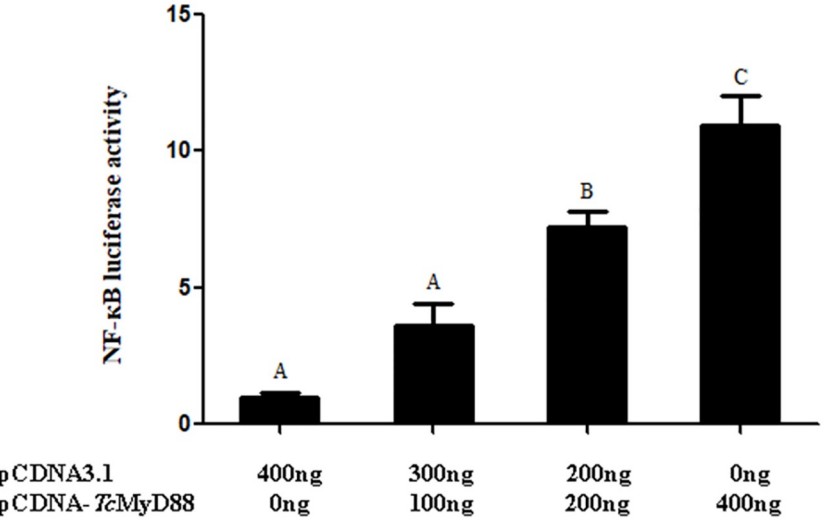

**Fig 7. Changes in the relative luciferase activity of NF-κB were analyzed based on levels of *Tc*MyD88 overexpression.** Significant differences are indicated by different letters.

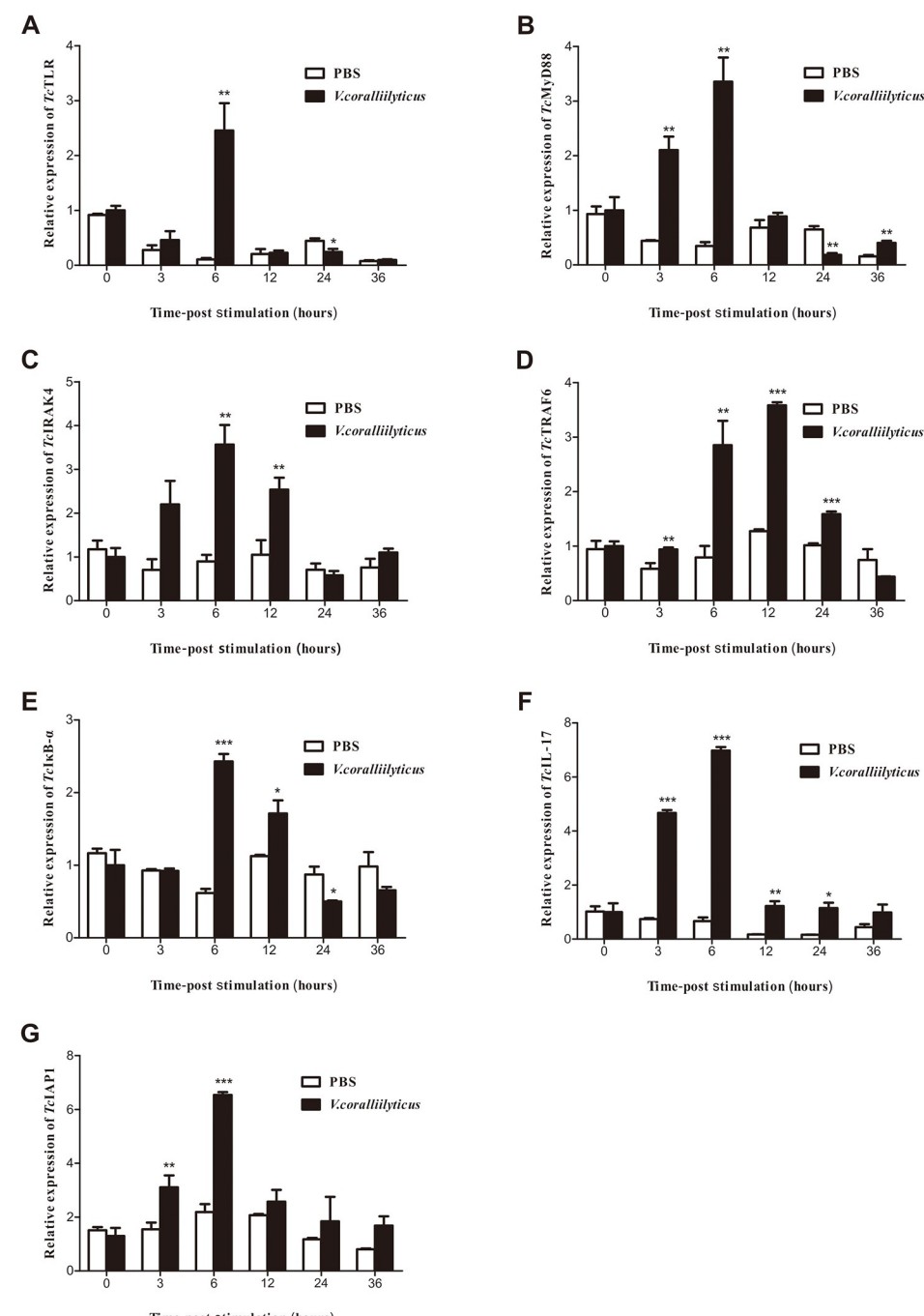

**Fig 8. Time-course expression analysis of TLR pathway-related genes in hemocytes from *V. coralliilyticus*-challenged *T. crocea*.** Expression profiles of TLR (A), MyD88 (B), IRAK4 (C), TRAF6 (D), IκB-α (E), IL-17 (F) and IAP1 (G) in hemocytes from *V. coralliilyticus*-challenged *T. crocea*. All of the samples were analyzed in triplicate. The β-actin gene was used as a reference gene to normalize expression levels between the samples. Statistical significance was determined by Student's t-tests and is indicated by an asterisk ($^{*}p < 0.05$, $^{**}p < 0.01$ and $^{***}p < 0.001$).

[49, 50]. While sequencing the genomes of *V. coralliilyticus* isolates has indicated a varying repertoire of potential virulence factors that may function independently or in concert to induce pathogenicity, hemolysin and extracellular protease activities are thought to play important roles during pathogenesis in oysters [47, 51, 52]. It was found that lytic enzymes produced by aquaculture pathogens include haemolysins and proteases [53]. Extracellular metalloproteases facilitate bacterial invasion and the infection process, acting to enhance tissue permeability and leading to necrotic tissue damage and cytotoxicity in the host [54]. The discovery in our study filled the research blank of rapid lethality caused by *V. coralliilyticus* in adult bivalves. Furthermore, we dissected this phenomenon from a mechanistic molecular level.

To shed light on the molecular response mechanism of *T. crocea* during *V. coralliilyticus* infection, we used transcriptome sequencing analysis and other relevant techniques to analyze the *T. crocea* hemocytes at 0 h, 6 h, 12 h and 24 h after *V. coralliilyticus* challenge. *T. crocea* employs the innate immune response as the sole defense mechanism against pathogen infection, such as pathogen recognition and apoptosis systems [55, 56]. Our results revealed that the total number of unigenes and DEGs was 195651 and 3446, respectively. More details were uncovered by KEGG pathway enrichment analysis, where DEGs were found to be significantly enriched in immune-related signaling pathways, such as the TLR signaling pathway and the NF-κB pathway, and some were associated with apoptotic pathways, such as the TNF signaling pathway. These results were consistent with the quantitative validation, where it was preliminarily demonstrated that the expression of TLR pathway-related genes changed after *V. coralliilyticus* challenge, and downstream genes were activated, which promoted the release of inflammatory factors such as IL-17 and led to the activation of a series of signals. Taken together, these results coincided with previous work in Pacific oyster larvae and adult *Nematostella vectensis* in which the TLR-to-NF-κB pathway was activated during pathogenic conditions [49, 57]. However, this activation reaction is not absolute, as shown here; some downregulated genes, such as the partially lectin family, are components of the complement system. The complement system enables one of the major innate immune mechanisms, which has the ability to remove microbes and attack pathogens, and the existence of a potential multicomponent complement system has been identified in shellfish [58]. When *C. gigas* was infected with *V. splendidus*, suppressed genes may have helped the bacteria escape the giant clam response, and afterward, the bacteria were able to establish the pathogenic intracellular and intravesicular forms mediated by localization that enhanced bacterial protection from an innate immune attack [59–61].

Furthermore, *Tc*MyD88 of the TLR pathway was identified by KEGG pathway enrichment analysis. MyD88 has been demonstrated to be a key adapter protein in TLR signal transduction that triggers downstream cascades involved in innate immunity in organisms ranging from mammals to mollusks, that includes human, fish, Drosophila, and *C. gigas* [62–66]. The MyD88-dependent pathway leads to the activation of NF-κB and the expression of proinflammatory genes, such as TNF and IL-1 [67, 68]. *Tc*MyD88 was found to have two conserved domains, the death domain and the TIR domain. Furthermore, the expression profile of *Tc*MyD88 and the activation of NF-κB both revealed that *Tc*MyD88 plays a crucial role in the regulation of the *T. crocea* immune system. These results verified the existence and significance of a MyD88-dependent signaling pathway in *T. crocea*. Acting as an intermediate receptor for signal transduction, MyD88 participates in the transmission of multiple signaling pathways. TNF expression and apoptosis have been reported to share the same signal transduction molecule, MyD88, in human myelomonocytic cells [69]. Meanwhile, the TLR pathway and other signaling pathways may increase the activation of MyD88 [70]. Therefore, *Tc*MyD88 may serve as a link among different immune regulatory mechanisms against *V. coralliilyticus* infection.

In conclusion, we discovered that *V. coralliilyticus* could cause acute mortality of adult *T. crocea* at 28˚C, which is the first evidence of the rapid lethality of *V. coralliilyticus* in adult bivalves at natural and agricultural temperatures, since previous work was mainly focused on mortality of shellfish larvae. Moreover, transcriptomic analyses of the differences in molecular mechanisms between healthy and infected giant clams were obtained, and abundant differentially expressed immune-related genes and signaling pathways were identified, which drew our attention to the TLR pathway. Following quantitative validation and functional analysis, the results suggested that adult *T. crocea* could initiate the innate immune response through the TLR pathway against Vibrio infection, where changes in TLR pathway-related gene expression promoted the release of inflammatory factors such as IL-17, leading to the activation of a series of signals driving activities such as apoptosis. Despite the fact that *V. coralliilyticus* appears to be a global bivalve pathogen, limited information about its pathogenicity, infection mechanism and disease mitigation is available. Further studies are needed on the immune defense mechanisms of adult giant clams. These studies will be conducive to the development of health management in aquaculture.

## Supporting information

**S1 Fig. Species distribution of the BLASTX matches of the transcriptome unigenes.** This figure shows the species distribution of unigene BLASTX matches against the NR protein database with a cut-off value $E < 10^{-5}$ and the proportions for each species. Different colors represent different species.
(TIF)

**S2 Fig. Unigenes from the four hemocytes samples (V0, V6, V12, V24) were annotated by GO.** Unigenes were annotated in three categories: biological processes (red), cellular components (green), molecular functions (blue). Each bar represents the relative abundance of unigenes classified under each category at level 2.
(TIF)

**S3 Fig. KEGG classification of *T. crocea* unigenes.**
(TIF)

**S4 Fig. Nucleotide sequence of MyD88 cDNA in *T. crocea*.** The predicted amino acids sequence. was shown below the nucleotide sequence. The start codon and stop codon were presented in red. The death domain and TIR domain were marked with gray shadow and black frame respectively.
(TIF)

**S1 Table. Primer nucleotide sequences used in this study.**
(XLSX)

**S2 Table. Differentially expressed genes in the comparison of control and injected samples.**
(XLSX)

**S3 Table. The information of differentially expressed genes in heat map.**
(XLSX)

## Acknowledgments

We would like to acknowledge the contributions of several other people to this work. We would like to thank Chuanjie Fu, Kunna Liu and Yunqing Li for their helps with samples collection. We appreciate the sequencing services of Novogene (Beijing, China).

## Author Contributions

**Conceptualization:** Xiang Zhiming, Ziniu Yu.

**Funding acquisition:** Yang Zhang, Yuehuan Zhang, Jun li, Shu Xiao, Haitao Ma.

**Methodology:** Yue Lin.

**Project administration:** Fan Mao.

**Resources:** Zehui Zhao, Zihua Zhou.

**Software:** Xiangyu Zhang.

**Writing – original draft:** Duo Xu.

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
