## [Decision Letter · Decision Letter 0]

22 Jan 2020

PONE-D-19-35528

Mechanistic molecular responses of the giant clam Tridacna crocea to Vibrio coralliilyticus challenge

PLOS ONE

Dear Dr zhiming,

Thank you for submitting your manuscript to PLOS ONE. After careful consideration, we feel that it has merit but does not fully meet PLOS ONE’s publication criteria as it currently stands. Therefore, we invite you to submit a revised version of the manuscript that addresses the points raised during the review process.

1) Reviewer #1 raised numerous concerns about experimental details.  This should be addressed by making appropriate changes to the Material & Methods section and to the text of the Results / Figure legends where appropriate.  

2) The concerns about the apoptosis assay in Fig. 1 need to be addressed.  While the percentages of cells in the quadrants change upon infection, there is no actual "cell population" coming up which in annexin V positive (and hence early apoptotic) - it appears like a general no specific increase in fluorescence signal.  Also the is a considerable number of PI+ cells at t=0 h raising concerns about a large number of cells dying during the cell extraction and purification procedure.  Also these cells should likely be PI+ but Annexin V negative; thus there is an apparent issue with the compensation between the fluorescence channels in this experiment ( take a look a the sample plots of the vendor of the kit used: http://www.vazymebiotech.com/products_detail/productId=84.html). One important positive control in this context would be a treatment of the cell with an apoptosis inducing agent (e.g. etoposide).

3) The assembled transcriptome dataset should also be made publicly available such that a consistent gene annotation for this organism can be established for future references. 

4) The labeling in heat map shown in Figure 3 is not really helpful - instead of only referring to Cluster XXXX YYYYY, common names should be utilized that give a clear hint on the function of the genes being differentially expressed. 

5) The description of the results regarding the GO/ KEGG classification (lines 311-322) is currently meaningless, e.g. it is quite obvious (even without knowing anything about the data) that the majority of genes fall in to classes like "cellular processes", "metabolic processes", "cell", and "cell part".  What else should they fall in? I strongly urge the author to use their biological "common sense" to omit this non-useful information (that likely was just copid as the highest ranking scores from a bioinformatics software produces) and rather include information that actually provides some biological insight into the immune system of their model organism.  

We would appreciate receiving your revised manuscript by Mar 07 2020 11:59PM. To enhance the reproducibility of your results, we recommend that if applicable you deposit your laboratory protocols in protocols.io, where a protocol can be assigned its own identifier (DOI) such that it can be cited independently in the future. For instructions see: http://journals.plos.org/plosone/s/submission-guidelines#loc-laboratory-protocols

We look forward to receiving your revised manuscript.

Kind regards,

Sebastian D. Fugmann, Ph.D.

Academic Editor

PLOS ONE

Journal Requirements:

2. In your Methods section, please provide additional information regarding the permits you obtained for collection of Tridacna crocea. Please ensure you have included the full name of the authority that approved the animal collection and, if no permits or approvals were required, a brief statement explaining why.

Reviewers' comments:

Reviewer's Responses to Questions

**Comments to the Author**

1. Is the manuscript technically sound, and do the data support the conclusions?

Reviewer #1: Partly

2. Has the statistical analysis been performed appropriately and rigorously? 

Reviewer #1: Yes

3. Have the authors made all data underlying the findings in their manuscript fully available?

Reviewer #1: Yes

4. Is the manuscript presented in an intelligible fashion and written in standard English?

Reviewer #1: Yes

5. Review Comments to the Author

Reviewer #1: The manuscript presented here is of interest, well-written and presents new data.

My main concern is about the functional demonstration of apoptosis induction by V. coralliilyticus.

First, I am not confident in the chosen flow cytometry areas. Can I see your dead and apoptosis-induced controls? For me, PI+ cells should be separated at 102. With such area, I am quite afraid about your control hemocyte mortality (more than 10%?)

Secondly, to my opinion, one unique technic (on one sampling point) is not enough to demonstrate apoptosis. Can’t you perform TUNEL assay, caspase dosage, TEM, …?

In introduction section, I would appreciate to find more information on the huge diversity of TLR in marine bivalves.

Material and method sections lack information

- How many individuals were challenges?

- Did you check bacterial suspension purity and concentration?

- Did you analyze moribund animals to ensure V. coralliilyticus ‘imputability’ in mortality?

- What is the difference in the hemolymph sampling explained L138-139 and 141-142 ?

- If hemocytes sampled at 0, 3, 6, 12, 24 and 36h, can we see flow cytometry results on all sampling points?

- Were the RT-QPCR analyses (L440) and RNA-seq analyses performed on the same biological samples

- L150. Triplicate = technical replicates and not biological replicates if I correctly understand what you mean

- Can you precise the volume/weight of tissue for RNA extraction L177

- For RNA, 230 nm is also informative (L180)

- For qPCR, were the cDNA used diluted or pure? Can you precise it L241?

In discussion section, I would recommend to authors to be more careful on the potential bacterial virulence factors that could induce the immune response measured (L485-500). There is a diversity of hemolysins for instance. Even if some hemolysins were described in other models as playing a role in apoptosis, you could not say that ‘some hemolysins […] in the secretome of V. cora […] may be responsible of this phenomena’ (L487). Are they expressed in vivo ? and which hemolysin are we talking about? Except by performing dual-RNAseq, you should be more moderate in this part of your discussion.

6. PLOS authors have the option to publish the peer review history of their article (what does this mean?). If published, this will include your full peer review and any attached files.

Reviewer #1: No

---

## [Author Response · Author response to Decision Letter 0]

3 Mar 2020

PONE-D-19-35528

Reviewer #1

1) Reviewer #1 raised numerous concerns about experimental details. This should be addressed by making appropriate changes to the Material & Methods section and to the text of the Results / Figure legends where appropriate. 

Response: We have appropriately changed the Material & Methods section as the reviewer suggested. We totally used 330 individuals to complete the experiment. Each experiment had biological repeats. Some imprecise statements have been corrected in the revised MS (please see line151-152, 287, 193). And the insufficient information has been improved and supplemented in the revised MS (please see line142-144, 190, 256). Through the reculture of bacteria from the moribund animals hemocytes and no deaths occurring after PBS and Vibrio alginolyticus injection, we can ensure V. coralliilyticus ‘imputability’ in mortality. Meanwhile, based on previous studies and our data, we chose the time point, 6 h to perform flow cytometry in consideration of the scarcity of experimental samples.

2) The concerns about the apoptosis assay in Fig. 1 need to be addressed. While the percentages of cells in the quadrants change upon infection, there is no actual "cell population" coming up which in annexin V positive (and hence early apoptotic) - it appears like a general no specific increase in fluorescence signal. Also the is a considerable number of PI+ cells at t=0 h raising concerns about a large number of cells dying during the cell extraction and purification procedure. Also these cells should likely be PI+ but Annexin V negative; thus there is an apparent issue with the compensation between the fluorescence channels in this experiment ( take a look a the sample plots of the vendor of the kit used: http://www.vazymebiotech.com/products_detail/productId=84.html). One important positive control in this context would be a treatment of the cell with an apoptosis inducing agent (e.g. etoposide).

Response: We have repeated the experiment of apoptosis as the reviewer suggested and submitted a new version of Fig 1, and we have compensated for the fluorescence channels appropriately according to the manufacturer’s instruction for both this time and before. Specific experimental operations and data processing were referred to Lin’s and Qin’s articles, where the early apoptosis of hemocytes in marine invertebrates may not be accompanied by the new “cell population” coming up [1-3]. Moreover, due to the specificity of species and the difficulty of experimental operation, some cells died during the previous experimental treatment. However, the control group and the experimental group were performed simultaneously, with the same protocol and statistical methods, which making the FACS results comparable and statistically significant differences credible. The Annexin-positive cells in the experimental group was significantly higher (P<0.05) than that in the control group, although there was no new “cell population” coming up, which indicates V. coralliilyticus infection could induce hemocytes apoptosis (please see the details below).

3) The assembled transcriptome dataset should also be made publicly available such that a consistent gene annotation for this organism can be established for future references. 

Response: We will release the assembled transcriptome dataset, as soon as the article is accepted.

4) The labeling in heat map shown in Figure 3 is not really helpful - instead of only referring to Cluster XXXX YYYYY, common names should be utilized that give a clear hint on the function of the genes being differentially expressed. 

Response: Thank you for reviewer’s helpful comments. We have revised the labeling in the heat map and submitted the new version of Fig 3.

5) The description of the results regarding the GO/ KEGG classification (lines 311-322) is currently meaningless, e.g. it is quite obvious (even without knowing anything about the data) that the majority of genes fall in to classes like "cellular processes", "metabolic processes", "cell", and "cell part". What else should they fall in? I strongly urge the author to use their biological "common sense" to omit this non-useful information (that likely was just copid as the highest ranking scores from a bioinformatics software produces) and rather include information that actually provides some biological insight into the immune system of their model organism. 

Response: Thank you for reviewer’s valuable comments and we changed the description of the results as reviewer suggested (please check the MS line 320-326, 336-342 and below).

Revised: GO classification is a unified gene functional classification system. The analysis of annotated transcripts demonstrated that 36361 matched unigenes (18.58%) were divided into 26, 20, 10 entries for three categories, biological processes (BP), cellular component (CC) and molecular function (MF), respectively. The unigenes in “binding,” “catalytic activity,” “signaling,” “response to stimulus,” “membrane,” which were potentially related to immune existed in higher percentages compared to counterparts at sub-categories at the whole transcriptome reference. They played important roles during the defense process after challenge. The results provided a comprehensive view for screening candidate genes related to immune and defense mechanism (S2 Fig).

In addition, we also used KEGG to identify the probable functional status of assembled transcripts. A total of 12912 unigenes (6.59%) were assigned to five main categories containing 232 KEGG pathways. Remarkably, immune system (558) was the well-represented term in the KEGG organismal systems category. Signal transduction (1455) was the most enriched terms in both environmental information processing and the whole transcriptome reference, suggesting the dominant position of various signal transduction in processing the stimulation of environment. In the metabolism category, abundant unigenes were found in the amino acid metabolism (510) sub-category. These results indicated that these unigenes may play a role in mediating V. coralliilyticus exposure and associated impacts (S3 Fig).

Reviewer #2

Reviewer #1: The manuscript presented here is of interest, well-written and presents new data.

1) My main concern is about the functional demonstration of apoptosis induction by V. coralliilyticus.

First, I am not confident in the chosen flow cytometry areas. Can I see your dead and apoptosis-induced controls? For me, PI+ cells should be separated at 102. With such area, I am quite afraid about your control hemocyte mortality (more than 10%?)

Secondly, to my opinion, one unique technic (on one sampling point) is not enough to demonstrate apoptosis. Can’t you perform TUNEL assay, caspase dosage, TEM, …? 

Response: Firstly, we have repeated the experiment of apoptosis as reviewer suggested and submitted the new version of Fig 1, where the control hemocyte (PBS injection group) mortality were all less than 10% (please see the details below).

Secondly, RNA-seq and RT-QPCR have confirmed the change of apoptosis-related genes expression. Moreover, it was at the same time point as flow cytometry, which further proves the occurrence of apoptosis.

2) In introduction section, I would appreciate to find more information on the huge diversity of TLR in marine bivalves.

Response: Thank you for reviewer’s valuable comments and we supplemented more information on the huge diversity of TLR in marine bivalves in the introduction (please check the MS line 108-117 and below).

Revised: Since the first TLR was identified in Drosophila melanogaster, large members of TLR family have been recently investigated in marine bivalves, such as Chlamys farreri, Chlamys nobilis, Mizuhopecten yessoensis, C. gigas, C. virginica, Mytilus edulis and Hyriopsis cumingii [30-36]. Among them, the innate immune regulation of TLR genes in bivalves has been reported in C. farreri (CfToll-1) and C. nobilis (CnTLR-1), respectively, both of which might be involved in the immune response against pathogen invasion [31, 37]. In addition, 23 TLRs were identified and arranged in 4 clusters according to extra-cellular LRR domain content in Mytilus galloprovincialis [38]. Furthermore, there were 83 TLR genes in the genome of C. gigas, 19 of which had different responses to Vibrio infection [39].

3) Material and method sections lack information

- How many individuals were challenges? 

Response: 330 individuals were challenged. 40 individuals were performed for flow cytometry considering individual differences and death. 180 were used to investigate mortality where the experiment was repeated three times, with 20 individuals in each group tested every time. Another 110 individuals were for RNA-seq and RT-QPCR, where 5 individuals were randomly sampled from each group at every time point and 50 of them were used to make up for the loss of death.

- Did you check bacterial suspension purity and concentration? 

Response: We have supplemented the information in the revised MS (please see line 142-144). Specifically, for bacterial challenge, purified bacterial inoculum were centrifuged for 10 min to collect bacterial pullets and were washed 3 times with PBS and re-suspended in 2×PBS to a concentration of OD600nm = 1.0.

- Did you analyze moribund animals to ensure V. coralliilyticus ‘imputability’ in mortality? 

Response: Hemocytes were collected from moribund animals and then homogenized and diluted in PBS and spread on Thiosulfate Citrate Bile Salts Sucrose (TCBS) Agar plates, followed by overnight culture. Through the successful reculture of bacteria, we can ensure V. coralliilyticus ‘imputability’ in mortality (please see the details below). 

Meanwhile, when giant clams were injected with an equal amount of PBS and Vibrio alginolyticus under the same conditions, there was no observable death, which also demonstrated that V. coralliilyticus should be the main cause of mortality in T. crocea.

- What is the difference in the hemolymph sampling explained L138-139 and 141-142 ?

Response: We have corrected the misunderstanding statement in the revised MS (please see line151-152). There is no difference in the hemolymph sampling explained L138-139 and 141-142.

Revised: Hemocytes were collected at scheduled intervals (0, 3, 6, 12, 24, and 36 h post injection) from both the challenged and control groups. Among them, hemocytes taken at 0, 6, 12, and 24 h post challenge were stored in liquid nitrogen for transcriptome analysis.

- If hemocytes sampled at 0, 3, 6, 12, 24 and 36h, can we see flow cytometry results on all sampling points? 

Response: It has been reported that after 6 h post-injection of V. coralliilyticus, haemocyte cell concentrations in haemolymph of infected and non-infected mussels were similar. However, the proportion of viable haemocytes in haemolymph of infected mussels was substantially lower than that in non-infected mussels [4]. Hence, we speculated that 6 h might be a critical time point. Meanwhile, in terms of phenotype, dead individuals appeared at 6 h post infection. Furthermore, in terms of mechanism, the expression of apoptosis-related genes like IAP1 was dramatically increased at 6 h, which both RNA-seq and RT-QPCR confirmed. Based on these, it is reliable that flow cytometry was performed at 6 h post infection. On the other hand, giant clams are rare and difficult to collect. Taking into account the loss of death, this experiment needs a large number of samples, which should be considered very carefully and ecological friendly.

- Were the RT-QPCR analyses (L440) and RNA-seq analyses performed on the same biological samples 

Response: The RT-QPCR analyses and RNA-seq analyses were performed on the same biological samples. Each sample was divided into two parts, one for RNA-seq, the other for RT-QPCR.

- L150. Triplicate = technical replicates and not biological replicates if I correctly understand what you mean

Response: We have corrected the misunderstanding statement in the revised MS (please see line287). We performed biological replicates and n=3.

- Can you precise the volume/weight of tissue for RNA extraction L177

Response: We have supplemented the information in the revised MS (please see line190). We sampled 50mg of each tissue for RNA extraction.

- For RNA, 230 nm is also informative (L180) 

Response: We have corrected the unspecific statement in the revised MS (please see line193).

Revised: The concentration and purity were examined at 260/230 and 260/280 absorbance ratios.

- For qPCR, were the cDNA used diluted or pure? Can you precise it L241? 

Response: We have supplemented the information in the revised MS (please see line256). For qPCR, all the template cDNA were diluted to 200ng/μl.

4) In discussion section, I would recommend to authors to be more careful on the potential bacterial virulence factors that could induce the immune response measured (L485-500). There is a diversity of hemolysins for instance. Even if some hemolysins were described in other models as playing a role in apoptosis, you could not say that ‘some hemolysins […] in the secretome of V. cora […] may be responsible of this phenomena’ (L487). Are they expressed in vivo ? and which hemolysin are we talking about? Except by performing dual-RNAseq, you should be more moderate in this part of your discussion. 

Response: Thank you for reviewer’s valuable comments and we have corrected the related description in the discussion (please check the MS line 510-514 and below).

Revised: Diverse metalloprotease and effector genes like the pore forming toxin hlyA were identified in the genome of V. coralliilyticus and expressed proteases were also detected in the secretome, which caused mortality in Drosophila and Artemia and may be involved in the infection of T. crocea.

References

1. Lin Y, Mao F, Zhang X, Xu D, He Z, Li J, et al. TRAF6 suppresses the apoptosis of hemocytes by activating pellino in Crassostrea hongkongensis. Dev Comp Immunol. 2020;103:103501. doi: 10.1016/j.dci.2019.103501. PubMed PMID: 31634519.

2. Qin Y, Zhang Y, Li X, Noor Z, Li J, Zhou Z, et al. Characterization and functional analysis of a caspase 3 gene: Evidence that ChCas 3 participates in the regulation of apoptosis in Crassostrea hongkongensis. Fish Shellfish Immunol. 2020;98:122-9. doi: 10.1016/j.fsi.2020.01.007. PubMed PMID: 31917320.

3. Wang X, Wang M, Xu J, Jia Z, Liu Z, Wang L, et al. Soluble adenylyl cyclase mediates mitochondrial pathway of apoptosis and ATP metabolism in oyster Crassostrea gigas exposed to elevated CO2. Fish & Shellfish Immunology. 2017;66:140-7. doi: 10.1016/j.fsi.2017.05.002.

4. Nguyen TV, Alfaro AC, Young T, Ravi S, Merien F. Metabolomics Study of Immune Responses of New Zealand Greenshell™ Mussels (Perna canaliculus) Infected with Pathogenic Vibrio sp. Marine Biotechnology. 2018;20(3):396-409. doi: 10.1007/s10126-018-9804-x.

---

## [Decision Letter · Decision Letter 1]

24 Mar 2020

Mechanistic molecular responses of the giant clam Tridacna crocea to Vibrio coralliilyticus challenge

PONE-D-19-35528R1

Dear Dr. zhiming,

We are pleased to inform you that your manuscript has been judged scientifically suitable for publication and will be formally accepted for publication once it complies with all outstanding technical requirements.

With kind regards,

Sebastian D. Fugmann, Ph.D.

Academic Editor

PLOS ONE

Additional Editor Comments (optional):

Reviewers' comments:

Reviewer's Responses to Questions

**Comments to the Author**

1. If the authors have adequately addressed your comments raised in a previous round of review and you feel that this manuscript is now acceptable for publication, you may indicate that here to bypass the “Comments to the Author” section, enter your conflict of interest statement in the “Confidential to Editor” section, and submit your "Accept" recommendation.

Reviewer #1: All comments have been addressed

2. Is the manuscript technically sound, and do the data support the conclusions?

Reviewer #1: Yes

3. Has the statistical analysis been performed appropriately and rigorously? 

Reviewer #1: Yes

4. Have the authors made all data underlying the findings in their manuscript fully available?

Reviewer #1: Yes

5. Is the manuscript presented in an intelligible fashion and written in standard English?

Reviewer #1: Yes

6. Review Comments to the Author

Reviewer #1: (No Response)

7. PLOS authors have the option to publish the peer review history of their article (what does this mean?). If published, this will include your full peer review and any attached files.

Reviewer #1: No

---

## [Editor Report · Acceptance letter]

26 Mar 2020

PONE-D-19-35528R1 

Mechanistic molecular responses of the giant clam *Tridacna crocea* to *Vibrio coralliilyticus* challenge 

Dear Dr. zhiming:

I am pleased to inform you that your manuscript has been deemed suitable for publication in PLOS ONE. Congratulations! Your manuscript is now with our production department. 

With kind regards,

on behalf of

Dr. Sebastian D. Fugmann 

Academic Editor

PLOS ONE